# A Complete Review of Mexican Plants with Teratogenic Effects

**DOI:** 10.3390/plants11131675

**Published:** 2022-06-24

**Authors:** Germán Chamorro-Cevallos, María Angélica Mojica-Villegas, Yuliana García-Martínez, Salud Pérez-Gutiérrez, Eduardo Madrigal-Santillán, Nancy Vargas-Mendoza, José A. Morales-González, José Melesio Cristóbal-Luna

**Affiliations:** 1Laboratorio de Toxicología Preclínica, Departamento de Farmacia, Escuela Nacional de Ciencias Biológicas, Instituto Politécnico Nacional, Av. Wilfrido Massieu 399, Col. Nueva Industrial Vallejo, Del. Gustavo A. Madero, Ciudad de México 07738, Mexico; gchamcev@yahoo.com.mx (G.C.-C.); moviangel13@yahoo.com.mx (M.A.M.-V.); 2Laboratorio de Neurofisiología, Departamento de Fisiología “Mauricio Russek”, Instituto Politécnico Nacional, Escuela Nacional de Ciencias Biológicas, Av. Wilfrido Massieu 399, Col. Nueva Industrial Vallejo, Del. Gustavo A. Madero, Ciudad de México 07738, Mexico; ygarciamart@hotmail.com; 3Departamento de Sistemas Biológicos, Universidad Autónoma Metropolitana-Xochimilco, Calzada del Hueso 1100, Del. Coyoacán, Ciudad de México 04960, Mexico; msperez@correo.xoc.uam.mx; 4Laboratorio de Medicina de Conservación, Escuela Superior de Medicina, Instituto Politécnico Nacional, Plan de San Luis y Díaz Mirón, Col. Casco de Santo Tomás, Del. Miguel Hidalgo, Ciudad de México 11340, Mexico; eomsmx@yahoo.com.mx (E.M.-S.); nvargasmendoza@gmail.com (N.V.-M.); jmorales101@yahoo.com.mx (J.A.M.-G.)

**Keywords:** traditional medicine, Mexican plants, alkaloids, pregnancy exposure, teratogenic effects

## Abstract

In Mexico, the use of medicinal plants is the first alternative to treat the diseases of the most economically vulnerable population. Therefore, this review offers a list of Mexican plants (native and introduced) with teratogenic effects and describes their main alterations, teratogenic compounds, and the models and doses used. Our results identified 63 species with teratogenic effects (19 native) and the main alterations that were found in the nervous system and axial skeleton, induced by compounds such as alkaloids, terpenes, and flavonoids. Additionally, a group of hallucinogenic plants rich in alkaloids employed by indigenous groups without teratogenic studies were identified. Our conclusion shows that several of the identified species are employed in Mexican traditional medicine and that the teratogenic species most distributed in Mexico are *Astragalus mollissimus*, *Astragalus lentiginosus*, and *Lupinus formosus*. Considering the total number of plants in Mexico (≈29,000 total vascular plants), to date, existing research in the area shows that Mexican plants with teratogenic effects represent ≈0.22% of the total species of these in the country. This indicates a clear need to intensify the evaluation of the teratogenic effect of Mexican plants.

## 1. Introduction

Since ancient times, humans have used the elements in their environment to satisfy their basic needs. Such is the case of plants, which for millennia have been utilized by humans to produce food, shelter, clothing, footwear, dyes and stains, means of transport, fertilizers, fragrances, cosmetics, as fuel and, of course, to alleviate their diseases [1,2]. There is solid evidence that plants have been cultivated for their biological effects for over 60,000 years [3], with the earliest written records on the preparation of herbal remedies and their biological effects being found in Sumerian tablets [4] and in Egyptian, Indian, and Chinese inscriptions aged approximately 5000 years, such as Ebers papyrus [5], Rigveda texts [6], and the book Pen T’Sao [7], respectively.

In places such as Greece and Central Asia, these first records have been found in more recent times, that is, approximately 2500 years ago [8], while in Mexico it is possible to find records of the use of plants with medicinal purposes (by the Olmec, Mayan, Mixtec, and Aztec), such as the *Códice De la Cruz Badiano* or the *Libellus de Medicinalibus Indorum Herbis* [9]. It is difficult to establish a concise period for the first records of their medicinal use, because the majority of Mesoamerican literature, in the form of *códices*, were burned in the years after the Conquest by Spanish missionaries [10]. In this regard, according to archaeological records, the process of domestication and the use of plants in Mesoamerica began about 7000 and 5000 years ago, respectively [11]. The latter can provide us with an idea of the important role that plants have played as a means of health in the native societies of the current territory of México.

### 1.1. From Plants to Drugs

According to the significance that Greece, the cradle of philosophical thought and Western civilization, has held in the sciences and arts, it should not come as a surprise to us that the Greeks were the fathers of Medicine, Botany, and Pharmacology (Hippocrates, Theophrastus, and Pedanius Dioscorides, respectively), who laid the foundations of the therapeutic value of medicinal plants through detailed compilations of the knowledge of medicinal plants during their respective times and in their respective regions [12,13,14], as well as that herbal medicine is denominated phytotherapy, a compound word formed by the Latin prefix *phyto* “plant”, and the Latin word *therapia*, “to treat medically”, in the study of the use of extracts of natural origin as medicines [15].

With the fall of the Western Roman Empire (476 AD), there began a period of time that stretched from the V to the XV century, called the “Middle Ages” or the “Dark Ages”, which involved a notable lag in the development of the sciences [16], despite that knowledge on the use of medicinal plants (and many others) survived during being confined for several centuries inside the walls of monasteries in countries such as England, Ireland, France, and Germany [17], among others. The Arab people were responsible for the preservation of a great part of Greco–Roman knowledge (along with that of the Chinese and the Indians, mainly in terms of plants), and later with the establishment of the first private pharmacies in Baghdad, Irak, at the end of the 8th century [18,19]. Later, the Persian pharmacist and poet Avicena (980–1037 AD) contributed to the dissemination of the knowledge of therapeutic plants with his work “*Canon Medicinae*”, considered the latest translation of all Greco–Roman Medicine [1] and the starting point for the development of medicinal-plant texts throughout Europe, such as “*The Corpus of Simples*” by Ibn al-Baitar, or the Florentine “*Nuovo Receptario Composito*”, which, alongside other manuscripts in England (1518), laid out the concept of “pure compounds” and promoted their development; a century later, these manuscripts would comprise the basis of the emergence of the first pharmacopoeia (First London Pharmacopoeia) [20]. Deriving from this idea, the first natural product to be marketed as isolated and pure was morphine, by Merck in 1826; while the first semi-synthetic medicine based on a natural product was aspirin by Bayer in 1899 [21].

The health needs of our increasing population have intensified the interest in developing more effective chemically synthesized compounds. In this context, since 1910, there was the “magic bullet” (Salvarsan or compound 606) by the German bacteriologist Paul Ehrlich [22], continuing all the way to modern times with the development of drugs used for chronic obstructive pulmonary disease, such as umeclidinium bromide [23]. Despite the good results obtained with chemical synthesis, the therapeutic potential of medicinal plants was not lost from sight. From plants such as meadowsweet (*Spiraea ulmaria*), poppy (*Papaver somniferum*), foxglove (*Digitalis purpurea*), and *barbasco (Lonchocarpus utilis*), drugs as remarkable as acetylsalicylic acid [24], morphine [25], digoxin [26], and diosgenin [27], respectively, have been obtained. Currently, this approach has become diversified thanks to the boom in novel processes for the extraction and identification of organic compounds, but mainly due to the increasing use of traditional medicine, which has been the starting point for information obtained throughout scientific research for the development of molecules that, under different conditions, would have been difficult or virtually impossible to conceive. In this regard, nature arises and is constituted, with chemical synthesis [28] and with biotechnology [29], one of the three main ways to obtain biologically active molecules today [30]. This said fact can be observed as reflected in the growing number of studies during the last decades that have provided valuable information to this scientific field, confirming the molecules present in the structures of numerous plants that possess a biological effect against diseases. However, in addition to the development of the knowledge of plant curative properties, the misconception of the safety of natural products has become a health problem that can be confirmed in several epidemiological and experimental assays.

### 1.2. Mexican Plants

Worldwide, Mexico has one of the richest diversities of plants composed of native and introduced species from various parts of the planet [31] that, together with their cultural wealth, constitute Mexican Traditional Medicine. In general terms, the concept of traditional medicine refers to a conventional denomination adopted by researchers to refer to empirical medical systems that are organized and based on various cultures of the world [32]. Thus, traditional medicine is made up of three main types according to the source of the remedy: that obtained from animal products; that of processed minerals, and Botany [33]. With regard to the latter, it is difficult to accurately determine the number of families, genera, or species of existing Mexican plants. Some reports estimate that Mexico has approximately 22,351 species of vascular plants (native and introduced) and calculate that at least 6500 species must be added to this calculation [34,35]. Of this total number (≈29,000 total vascular plants), it is suggested that between 3000 and 5000 plants are currently utilized with medicinal purposes in the country [36].

#### The Teratogenic Effects of Mexican Plants

During pregnancy, the organogenesis phase is the most critical period for generating birth defects, because it is at this time that the differentiation and specialization of tissues, structures, organs, and systems in the *conceptus* begin, with the grouping of cells in early patterns directed by gene expression toward specific sites in an organ [37]. Thus, during this period, it is most likely (but not exclusively so) that there occurs a multifactorial process known as teratogenesis. The latter, as a result of multiple interactions between environmental (physical, chemical, biological, and maternal diseases, clinical states, etc.) and endogenous (genetic background of the mother and the embryo/fetus) factors, can produce a wide range of congenital deformities in the developing fetus or in the newborn [38]. Although the mechanisms by which this process occurs are varied, the main ones include oxidative stress, folate antagonism, vascular disruption, neural crest cell disruption, endocrine disruption, and specific receptor- or enzyme-mediated actions, among others [39]. Therefore, the severity and type of alterations will depend, among other things, on the time of the interaction of the teratogenic agent with the embryo/fetus, as well as on the mechanism(s) involved in the disruptive process.

The wide diversity of plants in Mexico allows some of these to be used by humans to alleviate their diseases or as food [40,41], even for livestock. In Mexico, as in other developing countries, the use of medicinal plants remains the first health care available in many rural areas to alleviate the diseases of the most economically vulnerable population, without social security [42,43], as a substantially less expensive and affordable alternative to conventional therapies. Unfortunately, in this practice, quality is generally unproven and is solely based on the population’s beliefs. In contrast, insufficient attention is paid to their possible toxicity (unlike what happens with drugs), with the belief espoused for generations, e.g., “the natural remedy” is totally safe, without taking into consideration that plants could contain toxic compounds. Thus, although plants are undeniably an important source of health for vulnerable persons, it is important to make it known to the general population that “natural is not always safe”.

The data provided by several investigations demonstrate than an important number of herbs and herbal products have been implicated in poisoning [44], health problems [45], and alterations in embryonic/fetal development in humans [46] and animals [47]. This latter toxic effect of plants comprises an important problem worldwide, not only for public health, where birth defects are one of the main preventable causes of morbidity, mortality, and childhood disability [48], but also in the economic sector, since in different parts of the world, the consumption of teratogenic plants by pregnant females produces a wide variety of congenital anomalies in livestock, generating, in this manner, considerable losses of capital for companies and local ranchers [49].

This work provides an updated catalog of native and introduced vascular plants in Mexico with a teratogenic effect, in order to describe the main birth defects produced by these in different models, as well as their implications during pregnancy. In addition, we underscore those plants that should be avoided during the critical period of pregnancy because of their harmful effects on the developing embryo/fetus (abortifacient and/or teratogenic potential). Additionally, this review seeks to serve as the basis for generating novel research projects, new questions, a better understanding of the Mexican flora, and as a promoter of the responsible and safe use of medicinal plants. For example, in Mexico, there are a considerable number of plants termed “sacred” that are employed with religious purposes due to their hallucinogenic effects, plants that are rich in alkaloids (these will be discussed later), plants without (to our knowledge) studies that support their safety or that warn of their toxicity during pregnancy and that are currently used without any type of regulation by ethnic populations.

## 2. Methodology

A scoping review of Mexican plants with a teratogenic effect was conducted in the principal academic research databases (DOAJ, PubMed, ScienceDirect, Scopus, Web of Sciences and Springer) and academic search engines (Google Scholar, Science.gov, and Microsoft Academic). The search was carried out in the first instance by faceting the main ideas that make up our research, which were later condensed into keywords and which in turn were utilized in different combinations to feed the academic search engines and databases in order to identify potentially relevant studies for their inclusion in our review as follows: “teratogenesis, teratogen, teratogenic effect, teratogenic plants, plant extracts, sacred plants, Mexican plants, herbal, herbal medicine, medicinal plants, traditional Mexican medicine, natural products, phytotherapy, pregnancy, birth defects, and congenital malformations”.

To achieve this goal, we adopted as inclusion criteria, articles, papers, books, book chapters, reports, patents, and theses on plants native to or introduced into Mexico with proven teratogenic effects, published during the last 72 years (1950–2022) and, as exclusion criteria, studies that address another type of toxicity not related to our main objective, information on compounds or molecules other than those obtained from plants, information from unreliable sources, information from years prior to 1950, information that does not refer to the keywords raised, repeated teratogenic plants previously found, and works on teratogenic plants that are not present in Mexico. To perform this selection, we compared the plants in the studies found with the checklist of the native vascular plants of Mexico [34] and with the databases of the Invasive Species Compendium [50], the Herbario Virtual of the Comisión Nacional para el Conocimiento y Uso de la Biodiversidad [51], the platform iNaturalist (citizen science social network) [52], and the Red de Herbarios del Noroeste de México [53], to differentiate between native plants, plants introduced from other countries, and plants that do not present in Mexico.

Please note that this review was focused only on the variety of plants with teratogenic effects. Therefore, no information is included on the number of investigations conducted on the same plant by different authors, with different animal species, doses, or any other different variable.

## 3. Results

### 3.1. Teratogenic Plants from Mexico

In this review, 63 species of Mexican plants with teratogenic effects distributed in 56 genera were identified (Table 1). Compared to the enormous number of plant species in Mexico, this figure scarcely represents ≈0.21% of the total number of species in the country. In the studies consulted, the main plant parts employed were leaves, followed by seeds, fruits, roots, flowers, and bark. Evidently, this fact depended on the amount and type of the compound suspected as the teratogenic agent in the studied species. For example, alkaloids are mainly found in leaves, barks and, to a lesser extent, in roots [54], and cyanides are often found in rhizomes and, at lesser amounts, in fruits and seeds [55]. However, essential oils (a complex mixture of fatty acids, aldehydes, esters, phenols, ketones, alcohols, nitrogen, and Sulphur) are isolated from various plant structures, such as peel, stem, leaves, flowers, roots, and woods [56]. Although the majority of the studies were conducted on livestock, complete plants were administered to pregnant animals through their diet, mixed with food; in laboratory animals such as rats, mice, fish, frogs, and chickens, the majority of the investigations opted to extract the representative compounds of the plant-in-question, utilizing different types of solvents according to the compound that the plants were suspected to obtain; for example, aqueous extracts for flavonoids, alkaloids, steroids, terpenoids, phenols, and terpenes; methanolic extracts for sterols, flavonoids, some alkaloids, and lectins; ethanolic extracts for diterpenes, aldehydes, and polyketides, and organic extracts for lycopene, fatty acids, some alkaloids, sterols, and terpenes. As can be observed, the relative polarity of the solvents employed ranges from the lowest with hexane (0.009) and continues in increasing order with benzene (0.111), methylene chloride (0.309), ethanol (0.654), and methanol (0.762), to the compound with the highest relative polarity: water (1.0). That is, the authors played with the intensity of the interactions of solvents with their molecules-of-interest to extract apolar compounds by applying non-polar solvents, and vice versa [57].

The case of the alkaloids is very interesting because, as bases, they are scarcely soluble in water, but are very soluble in polar (alcohols) and apolar (ether, chloroform, hexane) organic solvents [58] Nevertheless, in the various studies in which they were identified, alkaloids were obtained through aqueous extracts and, in very few cases, by hydroalcoholic or organic extracts. This is most likely because the experiments involved attempted to imitate the way in which the plant is ingested by humans, in the form of teas (aqueous extracts).

The main compounds identified as the cause of the developmental alteration observed in different studies, in descending order, were as follows; alkaloids (22); unknown (20); terpenes (5); flavonoids (3); steroids (3); carbohydrates (2); cyanide (2) M lycopene (2); polyphenols (2); acids (1); aldehydes (1); amino acids (1); β-carotenes (1); fatty acids (1); lactones (1); gingerols (1); lectins (1); peroxides (1); polyketides (1), and sterols (1). It is not surprising that the majority of these investigations found alkaloids to be responsible for the teratogenic effect, as for decades there has been full knowledge that these compounds generate a wide variety of developmental defects in humans and animals [59]. However, due to the large number of studies in which the compound responsible for the teratogenic effect is not mentioned, leads us to think that this is because determining this data was not a priority for the study. It is curious that a wide variety of natural compounds with a low incidence of teratogenic effects was identified as teratogenic compounds. Thus, it is essential to consider these for future studies, as for several of these there are insufficient reports or citations in the literature specifying their teratogenic properties.

**Table 1 plants-11-01675-t001:** List of Mexican plants screened for their teratogenicity.

Teratogen						Animal/Model				
Plant	Used Part	Doses	Route of Administration	Compound	Mechanism	Species	Status	Malformations	Origin	Citation
*Acanthospermum hispidum*	Leaves, stalk, and seeds; AqEx	AqEx 10%, 0–600 mg/kg/day	IG, during organogenic period	Sesquiterpenoids (terpenoids) and phenolic compounds	-	Wistar rats	Normal	Anomalies in urinary system, palatosquisis, acampsia, ear heterotopic, cranial alterations, micrognathia, gastroschisis, extra ribs.	Introduced from Brazil	[60]
*Acanthus montanus*	Leaves; Met:MCEx (1:1)	1000 mg/kg	IG, GD6–15	β-sitosterol (sterol)	-	Wistar rats	Normal	Embryotoxicity, reduction in fetal body weight, crown-rump, tail lengths; reduced ossification of extremities bones.	Introduced from Africa	[61]
*Ageratum conyzoides*	Leaves; HAEx (80:20)	1000 mg/kg	Oral, GD17–20	Pyrrolizidine alkaloids	Oxidative stress	Wistar rats	Normal	Decrease fetal weight.	Native of Mexico	[62]
*Aloe barbadensis miller*	Leaf; juice	3 mL/rat	IG, GD4–15	-	-	Wistar rats	Normal	Reduction in fetal body weight, crown rump length, tail length; renal alterations as shrunken tubules, mild degeneration of glomeruli, narrowing of capsular spaces.	Introduced from Africa	[63]
*Alstonia macrophylla*	Leaves; EtEx	10 mg/mice	IG, GD8–15	Acetogenins (polyketides), styryl-lactones and alkaloids	-	ICR mice	Normal	Head shape angular, reduced elongation in the snout, abnormalities in digits of forelimbs and hindlimbs (separate fully or lack of some digits), altered body shape morphology, dead implants.	Introduced from Indonesia	[64]
*Alstonia scholaris*	Stem bark; HAEx (85:15)	480 mg/kg	IP, GD11	-	-	Swiss albino mice	Normal	Mortality, growth retardation, body length reduction, bent tails, syndactyly, delay in fur development, eye opening, pinna detachment, and vaginal opening, testes descent, ear unfolding.	Introduced from India	[65]
*Artemisia annua* L.	Leaves; OEx	70 mg/kg	IG, GD14–20	Artemisinin (sesquiterpene)	-	Wistar rats	Normal	Post-implantation losses.	Introduced from Asia	[66]
*Astragalus lentiginosus (locoweed)*	Leaves	25%	Oral, mixed with food during pregnancy	-	-	Columbia sheeps	Normal	Abortion, skeletal malformations (flexure of carpal joint, lateral rotation front legs, hypermobility hock and stifle joint), fetal edema, ascites, fetal hemorrhage, less gross effect on fetal cotyledons.	Native of Mexico	[67]
*Astragalus mollissimus Torr*	Leaves and stems	-	Oral, during pregnancy	Swainsonine alkaloids	-	Horses, lambs,	Normal	Skeletal defects, limb contractures.	Native of Mexico	[68]
*Astragalus pubentissimus*	Leaves and stems	-	Oral, mixed with food during pregnancy	Swainsonine alkaloids	-	Sheep and cattle	Normal	Abortions, lateral rotation of forelimbs, contracted tendons, anterior flexure, looseness of hock joints, flexure of the carpus, decreased length.	Introduced from North America	[69]
*Azadirachta indica*	Seeds oil	1.2 mL	IG, during whole pregnancy	-	Oxidative stress	Sprague Dawley rats	Normal	Anophthalmia, enlarged trachea, abnormally shaped sternebrae, macroglossia, exencephaly.	Introduced from India	[70]
*Cajanus cajan*	Leaves; AqEx	0–600 mg/kg/day	IG, during organogenic period	-	-	Wistar rats	Normal	Anomalies in urinary system, palatosquisis, acampsia, ear heterotopic, cranial alterations, micrognathia, gastroschisis, extra ribs.	Introduced from Brazil	[71]
*Cannabis sativa* L. *(ganga)*	Resin	4.2 mg/kg	IP, GD1–6	Cannabinoid alkaloids	-	Wistar rats; mice	Normal	Higher rate of resorptions and syndactyly, encephalocele, phocomelia; decrease in fetal weight and size; dental asymmetry.	Introduced from Asia	[72,73]
*Carthamus tinctorius*	Flowers; AqEx	1.2 mg/kg	IG, GD0–8	-	Cytotoxicity	Albino mice	Normal	Changes in external, internal and longitudinal diameters, open neuropore, changes in cellular orientation and cellular degeneration, disconnected lateral folds.	Introduced from Mediterranean	[74]
*Caryocar brasiliense*	Fruits (dried pulp); Oex	1000 mg/kg	IG, GD6–15	Oleic acid (fatty acid), β-carotene and lycopene	High antioxidant capacity, probably	Wistar rats	Normal	Bruises, developmental delay, distended abdomen, incomplete ossification, vertebra in dumbbell, sternebra misaligned or bipartite, ribs wavy or supernumerary, change in shape of bones, absent sternebra or ribs, shortening of forelimbs and hindlimbs.	Introduced from Brazil	[75]
*Cinnamomum zeylanicum*	Leaves; AqEx, Cex	70 mg/kg	IP, throughout pregnancy	-	-	Wistar rats	Normal	Fetal resorption.	Introduced from South Asia	[71]
*Conium maculatum*	Leaves, fruits	2 and 1 mL/day	IG; GD25–35 sheep and GD55–75 cows	Piperidine alkaloids (coniine, N-methyl-coniine, conhydrine, pseudoconhydrine, γ-coniceine)	Restricted fetal movement (sedative or anesthetic effect of alkaloids)	Columbia sheep, cows	Normal	Cleft palate, under-extension of carpals and pastern joints, limb and spinal deformities.	Introduced from Europe	[76,77]
*Cortex cinnamom* L. *(Lauraceae)*	Cortex; EtEx	0.5 mmol/embryo	In ovo	Cinnamaldehyde (aldehyde)	-	Chick	Whole embryo culture	Malformations and lethality.	Introduced from India	[78]
*Croton megalocarpus*	Leaves; MetEx	1000 mg/kg	IG, GD6–16	-	Interference with the mechanisms involved in the production or removal of amniotic fluid	Swiss albino mice	Normal	Fetal resorption, microcephaly, polyhydramnios.	Introduced from Africa	[79]
*Curcuma longa*	Rhizome; MetEx	125.0 μg/mL	In medium, 6 hpf	Catechin, epicatechin, and naringenin (flavonoids)	Weakened or damaged embryo protective layer (chorion)	Zebra fish	Whole embryo culture	Body deformities of larvae, kink tail, bend trunk, enlarged yolk sac edema.	Introduced from Southeast Asia	[80]
*Datura stramonium *	-	-	-	Unknown, possibly alkaloids	-	Pigs	Normal	Cleft palate, contracture-type skeletal defects.	Native of Mexico	[81]
*Derris elliptica*	Leaves; AqEx	0.50%	Culture medium	-	-	Zebra fish	Whole embryo culture	Reduced hatchability, lower heartbeat, delayed formation, undeveloped head and tail region, coagulation and death of embryos.	Introduced from India	[82]
*Descurainia sophia*	Leaves	10%	Oral, mixed with food during first 2 months of pregnancy	3-butenyl glucosinolate (glycosides)	-	Goats	Normal	Hypothyroidism, absence of hair, goiter, abnormally large birth weights.	Introduced from Eurasia	[83]
*Garcinia kola*	Seeds; AqEx	200 mg/kg	IG, GD1–5	Flavonoids	-	Sprague Dawley rats	Normal	Decrease in fetal weight, malformed left upper limb.	Introduced from Africa	[84]
*Ginseng (Spp)*	Root	780–1560 mg/kg	IG, throughout pregnancy	Ginsenoside (steroid glycosides and triterpene saponins)	Activation of ginseng saponins	Albino mice	Whole embryo culture	Malformation of sternum, defects in the lumbar vertebrae, bad union of transverse processes with vertebral body.	Introduced from China and Korea	[85]
*Indigofera spicata*	Seeds; ion-exchange extraction	1 mL/100 g (1 mL of extract equals to 10 g seeds)	IG, GD13	Canavanine (amino acid)	Amino acid antagonism	Sprague Dawley rats	Normal	Cleft secondary-palate, somatic dwarfism, hepatic toxic change.	Introduced from Africa	[86]
*Ipomoea carnea*	Leaves; AqEx	15.0 mg/kg	IG, GD5–21	Swainsonine (alkaloid) and caligestines	Inhibition of acidic/lysosomal ámannosidase and Golgi mannosidase II; and glicosydases	Wistar rats	Normal	Reduction in ossification centers, kidney symmetry, dilated renal pelvis, hemorrhagic kidney, dilated cerebral ventricle, hemorrhagic cerebrum, hemorrhagic thyroid gland, spongy lung, and embryotoxicity.	Native of Mexico	[87]
*Krameria cytisoides Cav*	Leaves; MetEx	1000 mg/kg	IG, GD6–15	Kramecyne (hexamer of cyclic peroxide monomers)	Inhibition of cyclooxygenase 2 and consequent inhibition of prostaglandin synthesis; and ROS imbalance due to an increase in antioxidant enzymes (most likely)	Wistar rats	Normal	Decrease of fetal length and weight, live fetuses; increase of post-implantation loss; incomplete ossification in cranial vault, pelvis, sternum; asymmetric sternebrae; rudimentary and undulate ribs.	Native of Mexico	[88]
*Lantana camara*	Leaves; HAEx (70:30)	7 g/kg	IG, 14 days prior to mating and over the pregnancy (GD21)	-	-	Wistar rats	Normal	Increase of resorption rate and post-implantation loss index, forelimbs poorly ossification, sternebra with incomplete ossification.	Native of Mexico	[89]
*Lavandula angustifolia*	Leaves; AqEx	15 mg/kg	IP, GD3–6	Linalool (terpene)	-	BALB/c mice	Normal	Limb displacement from symmetry axis, encephalocele, no brain development, scoliosis, hemorrhage, spinal cord protrusion, eye protrusion, lack of limb, ear, eye and exohepatic development.	Introduced from Mediterranean	[90]
*Lawsonia inermis*	Leaves; AqEx	10 mg/kg	IP, GD1–7	-	Decreased levels of progesterone and increased levels of estrogen	BALB/c mice	Normal	Abortion (fetal death).	Introduced from Africa	[91]
*Leucaena leucocephala*	Leaves	50%	Oral, mixed with food during pregnancy	Mimosine (alkaloid)	-	Goats	Normal	Abortions at different stages of pregnancy, congenital goiter.	Native of Mexico	[92]
*Luffa operculata*	Fruits; AqEx	4 mL/kg	IG, GD1–9	-	-	CD1 mice	Normal	Cleft palate, anencephaly, exophthalmia, exophthalmia, delayed bone development; and decrease in implantation sites, live fetuses, and birth rate.	Native of Mexico	[93]
*Lupinus caudatus*	Leaves and stems	-	Oral, mixed with food GD40–70	Anagyrine (alkaloid)	-	Cows	Normal	Multiple congenital contractures, torticollis, arthrogryposis, kyphosis, rib cage deformities, scoliosis.	Introduced from North America	[94]
*Lupinus formosus*	Whole plant	2.25–3.16 g/kg	IG, twice daily GD40–70	Ammodendrine, piperidine and quinolizidine alkaloids	Restricted fetal movement (sedative or anesthetic effect of alkaloids)	Cows	Normal	Cleft palate, spinal curvature, rib cage depression, and multiple congenital contractures involving limbs, spinal column, and neck.	Native of Mexico	[95]
*Lycopersicon esculentum*	Leaves	-	Oral, mixed with food	Solanidanes, spirosolanes alkaloids, glycoalkaloids, and lycopenes	-	Columbia sheep, cows	Normal	Brain defects and cleft palate.	Native of Mexico	[96]
*Manihot esculenta*	Rhizome	50 and 80%	Oral, mixed with food during GD0–15	Cyanide and cyanogenic compounds (linamarin)	-	Albino rats	Normal	Growth retardation, limb defects, microcephaly, open eyes, low fetal body weight, embryonic death.	Native of Mexico	[97]
*Mimosa tenuiflora*	Seeds	10%	Oral, mixed with food during GD6–21	Alkaloids	-	Wistar rats	Normal	Cleft palate, scoliosis, bifid sternum, aplasia of sternebraes, hypoplastic sternebrae, hypoplasia of ischium, femur, nasal bone, deformed occipital bone, microphthalmia, lordosis, a shorter head, and weight increased.	Native of Mexico	[98]
*Momordica charantia*	Fruits and seeds; AqEx	2 mL/rat	IG, GD7, 11 or 13	-	-	Sprague Dawley rats	Normal	Cryptorchidism, splenomegaly; bilateral testicular, hepatic atrophy; renal hypertrophy; splenomegaly; anencephaly and spinabifida.	Introduced from Africa	[99]
*Montanoa tomentosa*	Leaves; HAEx (70:30)	25 mg/kg	IG, GD14–21	-	Ischemia and oxytocic effect	Wistar rats	Normal	Fetal mortality, increased fetal body weight, and maternal bleeding.	Native of Mexico	[100]
*Morinda citrifolia*	Fruit; AqEx	300 mg/kg	IG, GD7–15	-	-	Wistar rats	Normal	Delayed ossification and variations in skull, vertebral column, ribs, forelimbs, hindlimbs, and sternum.	Introduced from Asia	[101]
*Moringa oleifera*	Leaves; AqEx	3000 ppm	In medium, at segmentation phase during 48 h	-	-	Zebra fish	Whole embryo culture	Absence or low heartbeat rate, growth retardation, yolk deformity, stunted tail, and embryotoxicity.	Introduced from India	[102]
*Nicotiana glauca*	Leaves, stems, flowers, and woody (contained 0.175–0.23% anabasine)	5–8 mg/kg	IG, twice a day GD32–41	Anabasine (alkaloid)	Reduction in fetal movement during by fetal pharmacologic neuromuscular blockade	Spanish-goats	Normal	Bilateral cleft palate, embryonic/fetal death, resorptions, maxillary hypoplasia, midfacial retrusion, contracture in spine, neck, and legs.	Introduced from Argentina and Bolivia	[103]
*Nicotiana humilis (plumbaginifolia)*	Leaves; MetEx	0.5 mg/kg	IG, throughout gestational period	Nicotinic alkaloids	-	Sprague Dawley rats	Normal	Increased the body weight, length; decreased tail length; dysplastic tail, curved tail; behavioral disturbance; delayed opening eyes, incisor eruption and hair appearance.	Native of Mexico	[104]
*Nicotiana tabacum*	Leaves and stalks; juice	-	IG, GD4–53	Nicotinic alkaloids	-	Duroc pig	Normal	Congenital limb deformities and contractures; arthrogryposis.	Introduced from Peru	[105]
*Oxytropis sericea*	Leaves and stems	-	Oral, mixed with food during pregnancy	Swainsonine alkaloids	-	Sheep and cattle	Normal	Abortions, lateral rotation of forelimbs, contracted tendons, anterior flexure, looseness of hock joints, flexure of the carpus, decreased length.	Introduced from North America	[69]
*Perovskia abrotanoides*	Flowers; EtEx (95%)	0.25 g/kg	IP, GD8–11	Tanshinones (abietane diterpene)	Cytotoxicity and apoptosis induction	CD1 mice	Normal	Resorption, stillborn, polydactyly, spina bifida, aglossia, exencephaly, hydrocephaly, tarsal extensor, gastroschisis, skeletal abnormalities, cranium anomaly, variation in vertebrae, ribs, sternum, pelvis and hind limbs.	Introduced from Asia	[106]
*Peumus boldus*	Leaves; HAEx	800 mg/kg	IG, GD7–12	Boldine (alkaloid)	-	Wistar rats	Normal	Absence of paw inferior, external ear, and tail; increase of resorptions; weight decrease.	Introduced from Chile	[107]
*Pinus ponderosa*	Pine needles; isolated acids	152 mg/kg	IG, twice daily GD250–252	Isocupressic acid	-	Cows	Normal	Abortion.	Native of Mexico	[108]
*Podophyllum peltatum*	Roots; AEx	Five times	Topic, 4 h at the end of 23rd, 24th, 25th, 28th, and 29th weeks of pregnancy	Podophyllotoxin (β-D-glucoside)	Probably by interference with cellular mitosis	Humans	Normal	Simian crease on the left hand and a preauricular skin tag; polyneuritis, limb malformations, septal heart defects, and intrauterine death.	Introduced from North America	[109,110]
*Prunus serotina*	Fruits and leaves	-	Oral, ad libitum	Cyanide	-	Yorkshire pig	Normal	No tail, no anus, very small external sex organs, hind legs plantarflexed below the hock (most were female).	Native of Mexico	[111]
*Rauwolfia vomitoria*	Leaves and roots; EtEx	250 mg/kg	IG, GD7–11	-	-	Wistar rats	Normal	Distortion of cardiac muscle nicleiand myocaridial fibers.	Introduced from Africa	[112]
*Ricinus communis-linn*	Seeds; MetEx	600 mg/kg	IG, GD1–12	Ricin and lectin	-	Wistar rats	Normal	Prevention of implantation, abortion, and significant reduction of fetal parameters; crown-rump length, tail length, and weigh.	Introduced from Africa	[113]
*Ruta chalepensis* L.	Leaves; AqEx	10 mg/kg	IP, GD9–17	-	-	Swiss Rockefeller mice	Normal	Increased frequency of fetal resorption, lower fetal weight, hemivertebra, mesocephaly, spina bifida and skeletal malformations.	Introduced from Mediterranean	[114]
*Ruta graveolens*	Leaves; AqEx	20%	Oral, mixed with water GD0–4	-	-	Swiss albino mice	Normal	Abnormal compacted and uncompacted morula, extruded blastomere, embryo transport slightly delayed, retarded embryonic development.	Introduced from southern Europe	[115]
*Senna alata*	Leaves; HEx	1000 mg/kg	IG, GD10–18	Alkaloids	-	Wistar rats	Normal	Fetal death, decrease of implantation sites and corpora lutea; increase of resorption index, pre- and post-implantation losses.	Introduced from South America	[116]
*Silybum marianum*	Seeds (probably); AqEx	200 mg/kg	IP, GD6–15	Silibinin (flavonolignans)	Inhibition of cyclooxygenase and apoptosis	BALB/c mice	Normal	Reduction of fetal body weight, and crown-rump length; growth retardation; limb malformations, mandibular hypoplasia, vertebral deviations in normal curvatures, kyphotic body, increased fetal resorption.	Introduced from Mediterranean	[117]
*Solanum tuberosum*	Isolated from root; α-chaconine and α-solanine	4.6 and 10.9 mg/L, respectively	In medium, during 96 h	Steroidal glycosides and alkaloids	Carbohydrate side chain attached to the 3-OH group of solanidine, appears to be an important factor in teratogenicity	Xenopus	Whole embryo culture	Growth inhibition, loose gut coiling (miscoiling), misshapen eyes, muscular kinking, microencephaly, lacking in facial and brain structures, mortality.	Introduced from Bolivia	[118]
*Treculia africana*	Outer coat fruit; HAEX (50%)	2.5 mg/kg	SC, GD6	Polyphenol	-	Wistar rats	Normal	Reduction of fetal body weight, hydrocephaly, anophthalmia, omphalocele, shift in position, bipartite vertrabral, rudimentary ribs, fused ribs, extra ribs.	Introduced from Africa	[119]
*Trigonella foenum graecum*	Seeds; AqEx	1000 mg/kg	IG, during entire period of pregnancy	Diosgenin (steroid) and alkaloids	-	Swiss albino mice	Normal	Decrease in litter size and in fetal body weight, aplasia of external ear, bump on the head, median cleft of lower lip, fetotoxicity.	Introduced from Asia	[120]
*Urtica dioica*	Leaves; MetEx	1000 mg/kg	Oral, GD6–16	-	-	Swiss albino mice	Normal	Fetal resorption.	Native of Mexico	[101]
*Veratrum californicum*	Root and top structures; Oex	200 g	Oral, twice on GD14	Alkaloids	-	Sheep	Normal	Cyclopia, fetal death.	Native of Mexico	[121]
* Zingiber officinale*	Rhizome; AqEx	50 g/L	Oral, mixed with water GD6–15	Gingerols and shaogoals	-	Sprague Dawley rats	Normal	Embryonic loss, advanced skeletal development, increased fetal weight.	Introduced from Asia	[122]

Information on the teratogenic effect of Mexican plants, details of the part used, type of extract, mechanism, animal model, and congenital anomalies found is presented. Alcoholic extract, AEx; aqueous extract, AqEx; chloroform extract, CEx; ethanolic extract, EtEx; hours post-fertilization, hpf; hexane extract, Hex; hydroalcoholic extract, HAEx; intragastric, IG; intravaginal, IVG; methanol/methylene chloride, Met: MCEx; methanolic extract, MetEx; oily extract, OEx; organic extract, Oex.

According to what was found in this review, the animal species employed for carrying out these investigations were very varied; according to their incidence in the publications, the preferred species was rat (Wistar 20, Sprague Dawley 6, albino 1), followed by mouse (Swiss albino 5, BALB/c 3, CD1 2, albino 2, ICR 1, Swiss Rockefeller 1, not specified 1), and in a few species such as sheep 6 (Columbia 3, not specified 3), cows (5 not specified), goats 3 (Spanish 1, not specified 2), pigs (Duroc 1, Yorkshire 1, not specified 1), fish (zebra 3) cattle (not specified 2), chicken (not specified 1), frog (Xenopus 1), horses (not specified 1), human (Homo sapiens 1), and lambs (not specified 1). The latter demonstrates one of our main premises for conducting this review: livestock is seriously affected by teratogenic plants. Similarly, the preferred route of treatment administration was intragastric by gavage (31), followed by orally mixed with food or water (16), intraperitoneally (8), in culture medium (5), subcutaneously (1), and topically (1). All were administered during the critical period of gestation (organogenesis) or during the most susceptible stages of pregnancy in order to obtain suspected alterations.

The majority of species used in the teratogenicity studies in this review are rat and mouse, which is not uncommon, because, for a long time, they have been utilized as the preferred species for scientific research due to being readily available, small in size, easy to handle, easy to maintain, they have a short life cycle, etc. This is not to mention that there is much accessible information on their very similar physiological, anatomical, and genetic characteristics to those of humans [123], a similarity that would be more difficult to obtain in other models, such as fish, birds, or amphibians. In this context, the remarkable anatomical and physiological similarities between animals (particularly mammals) and humans have made possible the development of studies in different areas to investigate the physiological processes, mechanisms of action, efficacy, and the toxicity of drugs in developmental animal models before applying these models to humans [124].

While in preclinical toxicology the use of animals is a highly appreciated resource due to their resemblance to humans, it is evident that not all of the results obtained in animals can be directly translated into a very complex organism. In organisms such as humans, the organs perform distinct highly integrated and regulated physiological functions that involve a complex network of hormones, circulating factors, and cross communication between cells in other compartments, to say the least. Therefore, before thinking about extrapolating to humans the results obtained in animal models, it is necessary to investigate these at all required levels in order to obtain a complete description and an understanding of the involved mechanisms. In this latter aspect, many authors consider that a predominant reason for the poor rate of translation in some studies between animals and humans lies not in the lack of similarity between the species, but in problems of internal validity in preclinical animal studies (poor study design, lack of measures to control bias, etc.) [125].

Since in toxicity tests it is possible to reproduce the toxic effects in different species, for several years investigators have sought an animal species that responds in the same manner as humans to intoxications, to establish a practical model that aids in efficiently extrapolating the toxic effects observed in animals toward humans. However, the conclusion reached was that no animal, not even the higher primates, responds precisely in the same way that humans do against toxic substances [126]. Variability among species is so significant that even within a species such as that of the human, we can find individuals of the same race who respond differently to the same substance/molecule [127]. However, as Litchfield [128] demonstrated, more than one half century ago by comparing 89 effects of 6 toxic substances on rat, dog, and human, the problem is not qualitative, but quantitative. The author concluded that when a toxic effect does not appear in any of the former two species (dog or rat), it will not appear in humans either. However, if the substance affects only one of these, it will also affect humans. Thus, these toxicity tests are not designed to prove that a compound is safe, but instead to characterize the effects that it can produce in animals to produce a series of considerations such as metabolic rate, size, anatomical compartments, pathological state, physiological barriers, among others, which allow for an estimation of the intensity of the expected effects on humans.

Herein we find the importance of the data collected in this review, in which we can observe that the majority of the studies were carried out in species that share physiological similarities with human, rat, and mouse (and only one of these in human). Therefore, according to what has already been stated, it is to be expected that the plants analyzed in this review (or their extracts), when ingested or exposed to developing humans, produce similar alterations (qualitative) to those generated in rat or mouse murine models, varying only in the intensity of said alteration (quantitative).

Although in our review only a sole case of teratogenicity in humans by plants was found, this does not imply that plants do not generate alterations in the development of humans. Simply, at the time of birth, various alterations can go unnoticed if the medical staff does not have adequate preparation, not to mention that the population that uses plants to relieve their ailments and illnesses are mostly low-income persons who do not usually present at medical care centers. In this way, the lack of evidence of teratogenicity in humans (clinical cases) in this bibliographic review due to the consumption of medicinal plants does not comprise evidence of their absence. However, it is evidence of the need to intensify research in this area.

Second, the developmental alterations found in both livestock and laboratory animals were varied to such a degree that their specific description would be unsuccessful. Therefore, we will only mention the most significant alterations produced by the main compounds. The main alterations produced by plants with alkaloids were a decrease in fetal weight and length, altered body-shape morphology, and contracted tendons; encephalocele, phocomelia, and cleft palate, abnormalities in digits, forelimbs, hindlimbs, skull, spinal deformities, a reduction in ossification centers, and a higher rate of resorption, embryotoxicity, and fetal death. For plants with terpenes, we find palatosquisis, acampsia, ear heterotopic, micrognathia, gastroschisis, encephalocele, polydactyly, scoliosis, hemorrhage, spinal cord protrusion, eye protrusion, exencephaly, hydrocephaly and anomalies in the urinary system, variation in vertebrae, ribs, sternum, pelvis, hind limbs and skull post-implantation losses, and resorption. For plants with flavonoids, these are as follows: body deformities; kink tail; bent trunk; limb malformations; mandibular hypoplasia; decrease in fetal weight and crown–rump length; malformed limbs; vertebral deviations, and increased and fetal resorption. For plants with cyanide, these are the following: growth retardation; limb defects; microcephaly; open eyes; low fetal body weight; no tail; no anus; very small external sex organs, and embryonic death. The exact mechanisms by which these alterations occurred have not been fully clarified. However, the consulted authors point out or suspect that the main causing mechanisms in the observed teratogenic effect include oxidative stress [62], ischemia [100], enzymatic inhibition [88,117], alteration in levels of estrogens [91], cytotoxicity [106], apoptosis [117], interference with cellular mitosis [109,110], and restricted fetal movement [76,77,95], among others. Obviously, such alterations can be produced by each one of these or by combinations of these mechanisms (Figure 1).

The last important data obtained from our review are the number of native and introduced teratogenic plants in Mexico. According to our results, of the 63 teratogenic species of Mexican plants found, 19 are native to the country while the 44 remaining plants were introduced during the last 5 centuries from Africa (10), Asia (10), South America (8), India (5), the Mediterranean region (4), North America (4), and Europe (3) (Figure 2). These data, in addition to exhibiting the botanical wealth of Mexico, also reveal to us an important part of the notable cultural influences that Mexico has received from the world. Note that, with respect to these 44 introduced species, many of their genera are present natively (with different species) in Mexico, including *Solanum* (143), *Croton* (127), *Astragalus* (92), *Lupinus* (80), *Senna* (62), *Indigofera* (33), *Cinnamomum* (23), *Artemisia* (11), *Nicotiana* (10), *Descurainia* (8), *Morinda* (3), *Garcinia* (2), and *Alstonia* (1) [34]. It is logical to suppose that many of these “not studied species”, by sharing the genus with species that have shown teratogenic effects in the laboratory, must share the same or similar families of metabolites with the ability to modify fetal/embryonic development.

In this regard, *Astragalus* and *Lupinus* are the more interesting genera due to their high number of species distributed in Mexico and the severe malformations that they generate in livestock. Additionally, although *Solanum* and *Croton* are two of the most important genera in Mexico due to their high variety of species, in our review no reports were found indicating that they have affected livestock (the teratogenic effect of these two genera were proven only in mouse and frog). This does not mean that these genera do not affect livestock, but simply that these effects have not yet been documented or studied, rendering novel opportunities.

### 3.2. Sacred Plants from Mexico

Additionally, it is important to consider that there are some species of Mexican plants that contain considerable amounts of the most common teratogenic compounds in nature, that is, alkaloids, but their teratogenic effect, to our knowledge, has not been investigated. Such is the case of *Borago officinalis* [134], the genus *Senecio* [135], and, of course, an important group of Mexican plants termed “sacred” (Table 2). In ancient Mexico (before the Spanish Conquest), plants with toxic properties were usually considered sacred, because their consumption induces mystical outbursts, fear, and alteration of the human mind in order to experience an elevation in consciousness; for this reason, they were bestowed with the personality of deities based on their effects [136]. In this manner, even today, the act of consuming *peyote* (*Lophophora williamsii*, America’s most famous sacred hallucinogen) is called by an indigenous group, the *Huicholes*, as receiving “*hikuri*,” which means receiving the “heart of the Deer God,” who in turn is known as *Tatewari* and represents the grandfather god (God of Fire). This deity, the original shaman, collects *peyote* every year and guides his followers on a mystical pilgrimage to a sacred place where their ancestors rest: *Wirikuta* [137].

The Mexican sacred plants mentioned in Table 2 possess an important role in rituals and in their medicinal use due to the presence of psychoactive alkaloids, such as psilocybin, mescaline, and ergotamine, among others [138]. However, an important case that we must specify here is that of *Salvia devious* or *hierba de la virgen*, which contains a powerful hallucinogenic substance called “salvinorin A,” with a structure similar to that of alkaloids. Nonetheless, it does not contain any nitrogen atom, and, thus, it is not considered as one of these, but rather the first documented diterpene nonalkaloid hallucinogen [139,140,141]. Despite the fact that *Salvia divinorum* is a sacred plant of great relevance in Mexico, it is addressed outside of Table 2 because, in said table, we find grouped only the Mexican sacred plants whose main hallucinogen components are alkaloids (molecules with great teratogenic capacity), while salvinorin A is a nonalkaloid hallucinogen whose teratogenic potential has not yet been evaluated or related to its particular structure.

The use of plants with ornamental, nutritional, aromatic, medicinal, and religious purposes is widely extended in Mexico [142]. In this respect, sacred plants with biological effects comprise an important element in indigenous medical and religious practices [143]. According to the lack of studies found in this review to evaluate the risks of these sacred plants for embryonic and fetal development, which are commonly used in Mexico, it is necessary to emphasize the danger of their use in the indigenous medical and religious systems. The fact is more alarming if we consider that about 80% of persons worldwide depend on traditional medicine to treat their ailments [144], and that, as stated by the Cámara de Diputados del Congreso de la Unión, the use of sacred plants in Mexico still continues not to be evaluated. Although it is true that Mexico, in subscribing to international treaties, is obliged to prohibit any substance decreed by the World Health Organization (WHO), there is no law in Mexico that punishes the use of these substances. In other words, Mexican legislation sanctions the possession, but not the consumption, of illegal substances, including sacred plants. Therefore, the consumption of hallucinogenic plants such as *peyote* is legal with only certain restrictions (trafficking) that do not apply to indigenous groups because “ot is an issue of rights regarding the culture and native communities of our country” [145].

**Table 2 plants-11-01675-t002:** List of Mexican sacred plants without studies of teratogenicity.

Scientific Name	Common Name	Compound	Type	Cite
*Ariocarpus retusus*	Peyote cimarron	Alkaloids	Cactaceae	[146]
*Argemone mexicana*	Chicalote	Alkaloids	Plat	[147]
*Conocybe*	Teonanácatl	Alkaloids	Mushroom	[148]
*Coryphantha compacta*	Biznaga Partida Compacta	Alkaloids	Cactaceae	[149]
*Datura ceratocaula Ortega*	Torna loco	Alkaloids	Plant	[150]
*Datura inoxia Mill*	Toloache	Alkaloids	Plant	[151]
*Echinocereus triglochidiatus*	Alicoche Copa de Vino	Alkaloids	Cactaceae	[152]
*Epithelantha micromeris*	Ikuli mulato	Alkaloids	Cactaceae	[153]
*Erytrhina americana Mill*	Zumpantle	Alkaloids	Plant	[154]
*Heimia salicifolia*	Sinicuichi	Alkaloids	Plant	[155]
*Ipomoea violacea*	Quiebra platos	Alkaloids	Plant	[156]
*Lophophora williamsii*	Peyote	Alkaloids	Cactaceae	[157]
*Lycoperdon mixtecorum*	Hongo de primera clase	Alkaloids	Mushroom	[158]
*Mammillaria senilis*	Biznaga cabeza de viejitos	Alkaloids	Cactaceae	[149]
*Pachycereus pecten-aboriginum*	Cardón Barbón	Alkaloids	Cactaceae	[159]
*Panaeolus sphinctrinus*	Toshka	Alkaloids	Mushroom	[160]
*Psilocybe caerulescens*	Quélet	Alkaloids	Mushroom	[160]
*Rhynchosia phaseoloides*	Ojo de cangrejo	Alkaloids	Plant	[136]
*Solandra brevicalyx*	Tecomaxóchitl	Alkaloids	Plant	[161]
*Sophora secundiflora*	Mezcal frijol	Alkaloids	Plant	[162]
*Stropharia cubensis*	San Isidro	Alkaloids	Mushroom	[163]
*Tagetes lucida*	Yauhtli	Alkaloids	Plant	[164]
*Turbina corymbose*	Ololiuhqui	Alkaloids	Plant	[156]
*Turnera Diffusa*	Damiana	Alkaloids	Plant	[165]
*Ungnadia speciosa*	Monilla	Alkaloids	Plant	[166]

Information is shown on the alkaloid-rich main Mexican sacred plants that do not, to our knowledge, have scientific studies that evaluate their toxicity on embryo–fetal development.

Finally, we would like to mention that it is advisable to develop a database on non-teratogenic plants, as it is difficult to determine which plants lack a teratogenic effect because, to our knowledge, there are no published studies reporting this. In other words, the corresponding studies may have been carried out, but due to the lack of a teratogenic effect, these resulted as being unattractive for their publication. This catalog of non-teratogenic plants would be very useful in various investigations, both experimental and theoretical.

## 4. Conclusions

The information compiled in this bibliographic review reveals that there are a considerable number of Mexican plants used in traditional medicine with proven teratogenic effects in laboratory animals and in livestock. The main substances responsible for the teratogenic effects of the plants studied were those of alkaloids, terpenes, and flavonoids. Among the most notable teratogenic effects, alterations to the nervous system, the axial skeleton, and specific systems/structures are highlighted, for which the alkaloids are those mostly responsible.

Many of these plant species share the genus and, most likely, the teratogenic effects with other Mexican plant species on which, to our knowledge, teratogenic studies have not been conducted. Of the 63 species of Mexican plants with a teratogenic effect, 30.15% are native to Mexico, while 69.84% derive from other regions such as Africa, Asia, and South America.

It was determined that there is a group of sacred Mexican plants, attractive due to the presence of alkaloids in their chemical composition and because of the use that they are afforded in traditional medicine and religious practices because of their hallucinogenic effects, which have, to our knowledge, no studies ascribed to them that describe their effects on development.

The biodiversity of plants in Mexico is very extensive: the 63 species with teratogenic effects found in this review barely refer to ≈0.21% of the total species in the country. Therefore, the need to continue exploring the teratogenic potential of Mexican plants is evident, at least beginning with the genera of other species that have already demonstrated their teratogenic effect in the laboratory.

## Figures and Tables

**Figure 1 plants-11-01675-f001:**
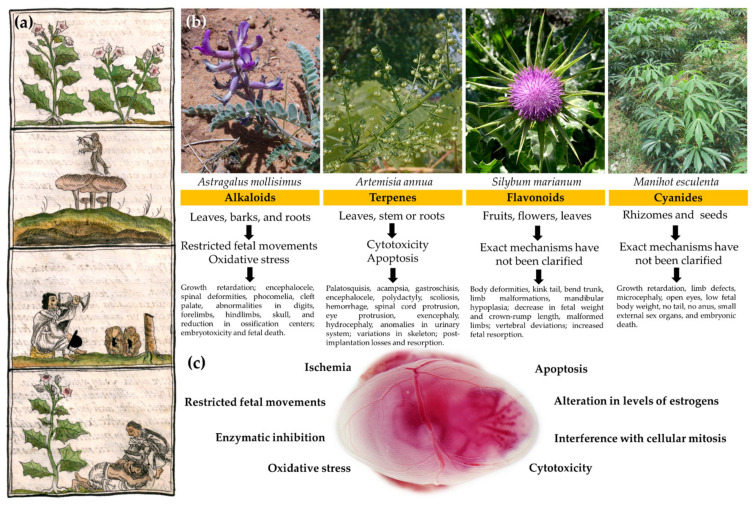
Developmental defects and the mechanisms of teratogenicity of the main compounds identified in Mexican plants. (**a**) In the upper part, this fragment of the Florentino codex shows the importance of the role that plants played in the pre-Hispanic Mexican culture (a reflection of the current one). Hallucinogenic plants, from top: *tlapatl, nanacatl, peyotl,* and *tolo* (bottom) used by the ancient indigenous people to stimulate the nervous system, for ritual or medicinal purposes as in love sickness [129]. (**b**) Four of the following teratogenic plants most used by the Mexican population as medicine or food are depicted: *Astragalus mollissimus* [130]; *Artemisia annua* [131]; *Silybum marianum* [132], and *Manihot esculenta* [133]. At the bottom of the fragment, we find the responsible compound, the part of the plant in which the compound predominates, the proposed mechanisms of teratogenicity, and the observed teratogenic effects. (**c**) A 12.5 GD mouse embryo is shown inside its amniotic sac to represent the teratogenic mechanisms identified in Mexican plants that can affect its development. Although the exact mechanisms have not been fully clarified, the authors suspect that the main mechanisms in the observed teratogenic effect are as follows: oxidative stress; ischemia; enzymatic inhibition; alteration in levels of estrogens; cytotoxicity; apoptosis; interference with cellular mitosis; restricted fetal movements, among others. It is noteworthy that such alterations can be produced not only by a single mechanism, but also by the interaction of several of these.

**Figure 2 plants-11-01675-f002:**
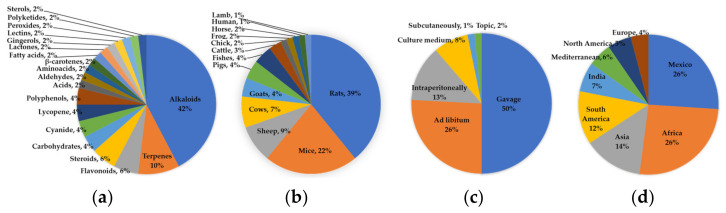
Graphic summary of the most relevant data obtained in the documentary research. (**a**) Abundance of the families of compounds identified in the teratogenic plants analyzed; (**b**) percentages of the main animal species (models) used to evaluate the teratogenicity of plants in the collected studies; (**c**) main routes of administration utilized in the evaluation of teratogenicity in the studies consulted, and; (**d**) distribution of identified teratogenic plants, native to or introduced from other countries.

## Data Availability

Not applicable.

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
