# Peer review of "A Complete Review of Mexican Plants with Teratogenic Effects"

_plants, 2022, doi:10.3390/plants11131675_

Round 1

Reviewer 1 Report

Dear, José Cristóbal-Luna, Ph.D.

Please check and revise.

-------------------------------------------------------------------------------------------------------------------------

I thought there was no revision.

-------------------------------------------------------------------------------------------------------------------------

Author Response

Comments from Reviewer 1

Comment 1: Extensive editing of English language and style required.

Response 1: Thank you for your observation. The corrected manuscript has been revised by a native English-speaking editor acquainted with the subject.

Comment 2: I thought there was no revision.

Response 2: The manuscript has been sent again for style revision, on this occasion, to a person associated with the subject of the manuscript. We hope that this time the syntax is adequate.

Reviewer 2 Report

This review reports a list of Mexican plants with teratogenic effects and describes their main alterations, teratogenic compounds, and the models and doses used. The topic is interesting and unusual. The cited references are appropriate. In my opinion the review can be accepted in this form for publication.

Author Response

Comments from Reviewer 2

Comment 1: Extensive editing of English language and style required.

Response 1: Thank you for your observation. The corrected manuscript has been revised by a native English-speaking editor acquainted with the subject.

Comment 2: This review reports a list of Mexican plants with teratogenic effects and describes their main alterations, teratogenic compounds, and the models and doses used. The topic is interesting and unusual. The cited references are appropriate. In my opinion the review can be accepted in this form for publication.

Response 2: Thank you for your comment. In effect, the objective of this review was to collect all of the relevant information that has been published in relation to the evaluation of the teratogenic effects derived from Mexican plants (species, structure used, compound, dose, models, routes of administration, etc.).

Reviewer 3 Report

The manuscript submitted by the authors is extremely interesting and could be of importance to be published in Plants. Teratogenicity of plants is often ignored. 

General recommendations:

However, the English is mostly unreadable. The authors should ask for a professional English editing service prior to resubmission.

Next, the structure of the Review is not suitable. As it is not a research paper, the structure IMRAD is redundant. The authors can place the REview methodology after the Indroduction (as section 2). and then The results can be given a more concrete name. Maybe the authors could split the section into more subsections. For instance: 3. Teratogenic plants from Mexico; 4. Sacred plants from Mexico. 5. Conclusions

Most of the studies in Table 1 are from animal studies. The authors should discuss somewhere how is this relevant or how does it transfer to humans? Are there any human studies or reports about the teratogenicity. Can we extrapolate the animal studies conclusions to humans? Please emphasize the lack of human studies.

I feel that the Review could benefit from a few comprehensive figurse that can make it more graphical and attractive to the Readers. One figure can sum up the teratogenic effects, with their putative mechanisms of actions (for example). Another figure can provide some graphics/pie charts with the statistical data that the authors are commenting on lines 200-300.

Line specific comments:

L21: ‘nervous system and axial’?? What do the authors mean with ‘axial’?

L25: Delete Torr or add for all the other species the taxonomist.

L27: ‘its study’? Whose study?

L87: Delete the comma after meadowsweet

L119: Replace ‘effect’ with ‘effects’

L161: Replace ‘pretends’ with a more suitable word

L169: There is a “T” at the very beginning

L191: Table 1, head row: Replace ‘specie’ with ‘species’

L221: Replace ‘specie’ with ‘species’

L284: Mention the botanical name of Peyote or the composition, if known

L298: Salvia divinorum should be italicized.

L301: It is not clear what is the reason why this plant is actually excluded from Table 2. Because it is non-alkaloid compound? The authors should be clearer.

L311: It would be good if the authors stress on which plants are restricted by the Mexican authorities, from those sacred or teratogenic!

L326-328: The key words should be placed in ‘’, as they were searched exactly as mentioned.

L335 Why information from years prior to 1971 was exclude, if above (L332) the authors claim that they looked in the last 71 years

Author Response

Comments from Reviewer 3

Comment 1: Extensive editing of English language and style required.

Response 1: Thank you for your observation. The corrected manuscript has been revised by a native English-speaking editor acquainted with the subject.

Comment 2: Next, the structure of the Review is not suitable. As it is not a research paper, the structure IMRAD is redundant. The authors can place the Review methodology after the Introduction (as section 2). and then The results can be given a more concrete name. Maybe the authors could split the section into more subsections. For instance: 3. Teratogenic plants from Mexico; 4. Sacred plants from Mexico. 5. Conclusions.

Response 2: Thank you for your insightful observations. After analyzing your recommendation, we agree that the structural adjustments you propose will make the content of this revision better organized, due to which all of your valuable suggestions were taken into account.

Comment 3: Most of the studies in Table 1 are from animal studies. The authors should discuss somewhere how is this relevant or how does it transfer to humans? Are there any human studies or reports about the teratogenicity. Can we extrapolate the animal studies conclusions to humans? Please emphasize the lack of human studies.

Response 3: Thank you for your suggestions. We consider that discussing these aspects in our results helps to point out the relevance of our review. Therefore, your suggestions were fully addressed in lines 275-331.

Comment 4: I feel that the Review could benefit from a few comprehensive figures that can make it more graphical and attractive to the Readers. One figure can sum up the teratogenic effects, with their putative mechanisms of actions (for example). Another figure can provide some graphics/pie charts with the statistical data that the authors are commenting on lines 200-300.

Response 4: Thank you for your recommendation. We think that including the information that you indicated graphically will aid us to improve the assimilation of the information, in that it as become more attractive and digestible this way. This point has been addressed in lines 356-372, and 395-401.

Comment 5: L21: ‘nervous system and axial’?? What do the authors mean with ‘axial’?

Response 5: Thank you for your question. It was, in fact, a typo; the text was intended to be "nervous system and axial skeleton". This has been corrected in line 21.

Comment 6: L25: Delete Torr or add for all the other speciese taxonomist.

Response 6: Thank you for your observation. It was corrected in line 26.

Comment 7: L27: ‘its study’? Whose study?

Response 7: Thank you for your question. Indeed, the sentence as written was not clear. Our intention was to compare the percentage of plants (with teratogenic effects) identified in our research, against the total number of plants in the country, in order to point out that, given the large number of plants in Mexico, there is a need to intensify the evaluation of the teratogenic effect of plants. The sentence has been corrected and restructured with the hope that it is now clearer. Lines 27-31.

Comment 8: L87: Delete the comma after meadowsweet

Response 8: Thank you for your observation. It was corrected in line 94.

Comment 9: L119: Replace ‘effect’ with ‘effects’

Response 9: Thank you for your observation. It was corrected in line 126.

Comment 10: L161: Replace ‘pretends’ with a more suitable word

Response 10: Thank you for your observation. It was corrected in line 164.

Comment 11: L169: There is a “T” at the very beginning

Response 11: Thank you for pointing out that typo. It was attended in line 177.

Comment 12: L191: Table 1, head row: Replace ‘specie’ with ‘species’

Response 12: Thank you for your insightful observation. This is now in Table, head row.

Comment 13: L221: Replace ‘specie’ with ‘species’

Response 13: Thank you for your observation. It was corrected in line 263.

Comment 14: L284: Mention the botanical name of Peyote or the composition, if known

Response 14: Thank you for your observation. It was included in line 412, mentioning the botanical name of peyote as “Lophophora williamsii”.

Comment 15: L298: Salvia divinorum should be italicized.

Response 15: Thank you for your observation. It was corrected in line 426.

Comment 16: L301: It is not clear what is the reason why this plant is actually excluded from Table 2. Because it is non-alkaloid compound? The authors should be clearer.

Response 16: Thank you for your question. In the text, it is now mentioned that Salvia divinorum was excluded from Table 2 because its hallucinogenic component is not strictly an alkaloid. In this regard, we think that your question is justified, since the sentence was not very clear. Thus, to be clearer, the sentence has been corrected and restructured with the hope that it is now improved.  Lines 430-434.

Comment 17: L311: It would be good if the authors stress on which plants are restricted by the Mexican authorities, from those sacred or teratogenic!

Response 17: Thank you for your observation. We find that addressing the point that sacred plants are prohibited or restricted to be a very interesting point. However, as mentioned, in Mexico there are no sanctions/prohibitions for the use of hallucinogenic plants. Existing sanctions are limited to their sale and trafficking. Therefore, to address your suggestion, in lines 444-451, we have delved deeper into this situation. We hope that it is now clearer.

Comment 18: L326-328: The key words should be placed in ‘’, as they were searched exactly as mentioned.

Response 18: Thank you for pointing this out. It has now been included in lines 184-187.

Comment 19: L335 Why information from years prior to 1971 was exclude, if above (L332) the authors claim that they looked in the last 71 years.

Response 19: Thank you for your question. It was, in fact, a typo; the text was intended to be "1950". Information prior to the last 72 years was excluded because, in a prior bibliographic review, it was determined that the oldest publications on the scientific evaluation of the teratogenic effect of plants dated from 1955. This has been corrected in line 193.

Round 2

Reviewer 3 Report

I congratulate the Authors for their efforts to improve significantly their work!